# Strong room-temperature bulk nonlinear Hall effect in a spin-valley locked Dirac material

Lujin Min [1,2], Hengxin Tan[3], Zhijian Xie[4], Leixin Miao[2], Ruoxi Zhang [1], Seng Huat Lee [1,5], Venkatraman Gopalan [2], Chao-Xing Liu [1], Nasim Alem [2], Binghai Yan [3] ✉ & Zhiqiang Mao [1,2,5] ✉

Nonlinear Hall effect (NLHE) is a new type of Hall effect with wide application prospects. Practical device applications require strong NLHE at room temperature (RT). However, previously reported NLHEs are all low-temperature phenomena except for the surface NLHE of TaIrTe$_4$. Bulk RT NLHE is highly desired due to its ability to generate large photocurrent. Here, we show the spin-valley locked Dirac state in BaMnSb$_2$ can generate a strong bulk NLHE at RT. In the microscale devices, we observe the typical signature of an intrinsic NLHE, i.e. the transverse Hall voltage quadratically scales with the longitudinal current as the current is applied to the Berry curvature dipole direction. Furthermore, we also demonstrate our nonlinear Hall device's functionality in wireless microwave detection and frequency doubling. These findings broaden the coupled spin and valley physics from 2D systems into a 3D system and lay a foundation for exploring bulk NLHE's applications.

The study of the Hall effect has led to important discoveries in condensed matter physics. For instance, the topological interpretation of the quantum Hall effect[1,2] inspired the research on topological phases of matters, which resulted in discoveries of various topological quantum states[3–10]. Recently, a new member of the Hall effect family, the nonlinear Hall effect (NLHE), has attracted a great deal of interest due to its fundamental importance and broad application prospect[11–19]. Unlike the conventional Hall effect or anomalous Hall effect (AHE) in the linear response region, which requires time-reversal symmetry breaking, the NLHE occurs under time-reversal symmetric conditions. NLHE is manifested by an alternating current (ac) driven second-harmonic Hall voltage response and a rectification effect; that is, a low frequency ($\omega$) ac current ($I^\omega$) generates not only an oscillating Hall voltage with a frequency of $2\omega$ ($V_\perp^{2\omega}$), but also a transverse rectified voltage component ($V_\perp^{RDC}$) at zero magnetic field, with both $V_\perp^{2\omega}$ and $V_\perp^{RDC}$ quadratically scaling with $I^\omega$. Recent theory predicts such quantum transport properties, if realized at room temperature, can be used

for high-sensitivity broadband Terahertz detection and thus enable Terahertz-band communication, which is believed to be a key technology for next-generation wireless networks[18]. Other potential applications of NLHE include low-power energy harvesting[20] and Néel vector detection[21,22].

Intrinsic NLHE arises from a topological band effect and is determined by the Berry curvature dipole (BCD), $\mathbf{D}_{ab} = \int \frac{d^n\mathbf{k}}{(2\pi)^n} \frac{\partial \Omega_{kb}}{\partial \mathbf{k}_a} f_0$ where $\Omega_\mathbf{k}$ is the Berry curvature, $n$ is the dimension, and $f_0$ is the Fermi distribution function[11,12]. $\Omega_\mathbf{k}$ describes local geometric properties of Bloch wavefunction, whereas the BCD reflects asymmetric distribution of $\Omega_\mathbf{k}$ in the momentum space. Non-centrosymmetric materials with Berry curvature hot spots near the Fermi level are generally expected to have finite BCD and can possibly show NLHE[11,13]. NLHE can also have an extrinsic origin such as skew scattering and side jump[17,23–25]. Intrinsic NLHE was first predicted[26,27] and demonstrated in bilayer WTe$_2$[13] and later in few-layer WTe$_2$[17]. Recently, several other 2D materials, including corrugated bilayer graphene[28], twisted bilayer WSe$_2$[29], and strained

[1]Department of Physics, Pennsylvania State University, University Park, PA, USA. [2]Department of Materials Science and Engineering, Pennsylvania State University, University Park, PA, USA. [3]Department of Condensed Matter Physics, Weizmann Institute of Science, Rehovot, Israel. [4]Department of Electrical and Computer Engineering, North Carolina Agriculture &Technical State University, Greensboro, NC, USA. [5]2D Crystal Consortium, Materials Research Institute, Pennsylvania State University, University Park, PA, USA. ✉e-mail: binghai.yan@weizmann.ac.il; zim1@psu.edu

WSe$_2$[30], were found to show NLHE. Besides these 2D materials, the surface states of the topological insulator Bi$_2$Se$_3$[31] and the Weyl semimetal TaIrTe$_4$[14] were also found to generate NLHE. Although theory predicts intrinsic NLHE can also occur in low-symmetry bulk crystals with tilted Dirac or Weyl points[24,26,27,32,33], bulk NLHE was well demonstrated only in three Weyl semimetals thus far, including Ce$_3$Bi$_4$Pd$_3$[34], T$_d$-MoTe$_2$, and T$_d$-WTe$_2$[35]. The NLHEs in 2D/3D materials mentioned above are primarily low-temperature phenomena. The only exception is TaIrTe$_4$, which exhibits NLHE at its surfaces in the 2−300 K temperature range[14]. Low-temperature NLHE apparently limits its applications. Bulk room-temperature NLHE, if realized, would facilitate exploration of NLHE's technological applications, but has not been reported yet.

In this work, we report a strong intrinsic NLHE observed near room temperature in bulk single crystals of a non-centrosymmetric Dirac material BaMnSb$_2$[36,37]. We observe not only ac-driven second-harmonic and rectification Hall responses but also a nonlinear Hall response driven by a direct current (dc). Moreover, our measurements also find the NLHE of BaMnSb$_2$ shows a maximum amplitude near room temperature, which agrees well with the calculated energy dependence of Berry curvature dipole, indicating the Dirac state origin of the observed NLHE. Lastly, we demonstrate wireless microwave detection and frequency doubling based on the observed NLHE.

## Results

### Theoretical prediction of the nonlinear Hall effect in BaMnSb$_2$

Previous work has demonstrated that BaMnSb$_2$ hosts a unique electronic state characterized by spin-valley locking[36,37], which is scarcely seen in bulk materials. This material possesses a layered, non-centrosymmetric orthorhombic structure with the space group of *Imm2* (No. 44), which is composed of alternating stacking of 2D Sb zig-zag chain layers and Ba-MnSb$_4$-Ba slabs, as shown in Fig. 1a, b. There are only two mirror planes preserved in this structure (denoted by $M_c$ and

$M_b$ in Fig. 1a, b). The lack of mirror symmetry along the *b*-axis leads to an in-plane polar axis parallel to the *a*-axis. Such a non-centrosymmetric orthorhombic structure generates a quasi-2D massive Dirac fermion state with two gapped Dirac cones emerging at K$^+$ and K$^−$ near the Fermi level ($E_F$), symmetrically located near the X-point along the X-M line, as illustrated in Fig. 1c. One distinct characteristic of such a Dirac state is the spin-valley locking which is enabled by the integration of the active valley degree of freedom with inversion symmetry breaking and strong spin-orbit coupling (SOC). Such a spin-valley locked Dirac state was evidenced by the angle-resolved photoemission spectroscopy and the spin-valley degeneracy of 2 extracted from the stacked quantum Hall effect[37]. Prior theoretical calculations also showed a small gap opens at the Dirac node in this system due to SOC[37,38].

According to the massive Dirac fermion model, one single pair of the spin-valley locked Dirac cones near $E_F$ would give rise to valley-contrasting Berry curvatures at K$^+$ and K$^−$[39]. Thus, a net BCD should be generated along the X-M line. To get quantitative information on the BCD of BaMnSb$_2$, we performed density-functional theory (DFT) band structure calculations and derived a realistic tight-binding model that fits the DFT results (see Methods). We should point out that the magnetism of Mn atoms has negligible effects on the energy bands near the Fermi surface. Therefore, we neglected the antiferromagnetic order in the following discussions and approximate that BaMnSb$_2$ has time-reversal symmetry. This is consistent with our experimental results of NLHE, which do not exhibit any transitions across the magnetic order temperature of 283 K[40]. As the Fermi surface of BaMnSb$_2$ is nearly 2D[37], a 2D tight-binding model was used to describe the band structure.

As depicted in Fig. 1d, the Dirac bands near the X-point exhibit large Berry curvature $\Omega_{xy}$ in the near-gap regime, and the two Dirac cones at K$^+$ and K$^−$ show opposite Berry curvature because $\Omega_{xy}$ is odd under the time-reversal symmetry. Furthermore, our calculations

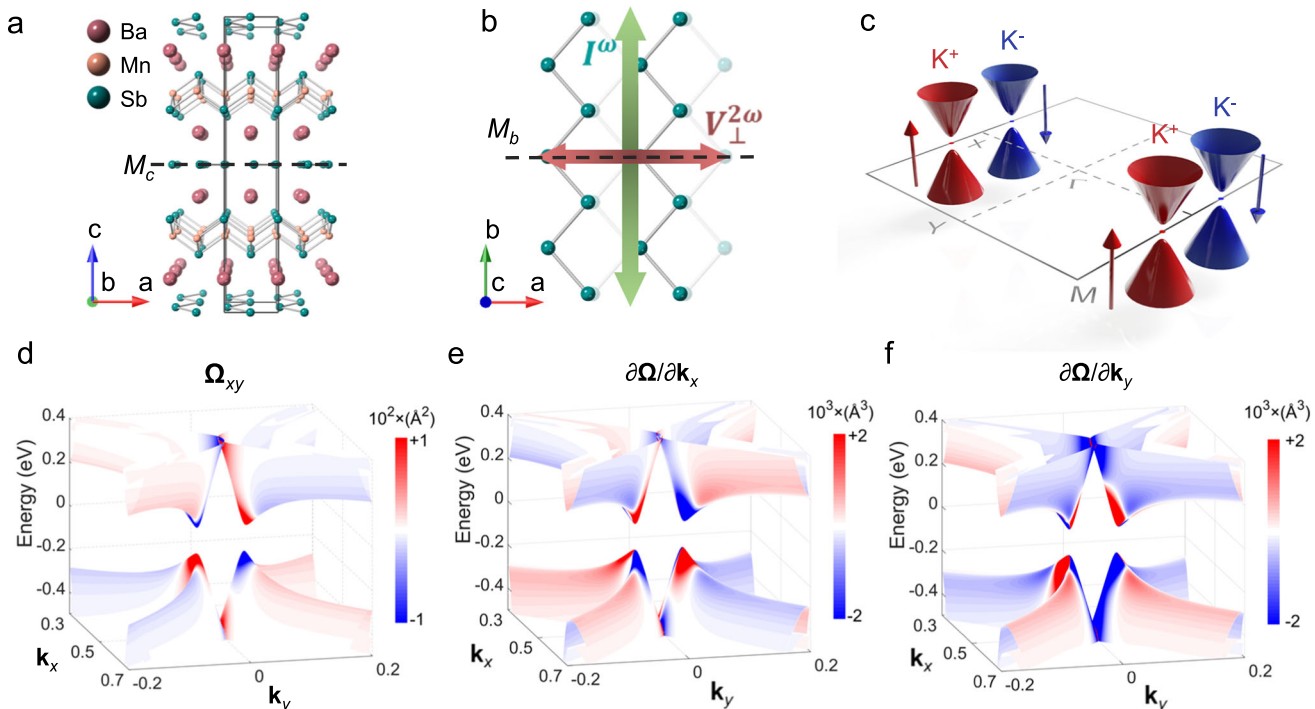

**Fig. 1 | Crystal structure, electronic structure, Berry curvature, and Berry curvature dipole distributions of BaMnSb$_2$. a** Crystal structure of BaMnSb$_2$. $M_c$ represents the horizontal mirror plane. **b** Top view of the Sb zig-zag chains in BaMnSb$_2$. The second Sb layer is shown in translucent color. $M_b$ represents the vertical mirror plane. This panel also schematically shows an ac $I^\omega$ flowing along the

*b*-axis generates a second-harmonic Hall voltage $V_\perp^{2\omega}$ response along the *a*-axis. **c** The schematic of the spin-valley locked Dirac cones located at K$^+$ and K$^−$ near the X-point, with the spin projection being color-coded (red, spin-up; blue, spin-down). **d**–**f** The color maps of the Berry curvature ($\Omega_{xy}$) and the first partial derivative of the Berry curvature ($\frac{\partial \Omega}{\partial k_x}$, $\frac{\partial \Omega}{\partial k_y}$) around the X-point where two massive Dirac cones exist.

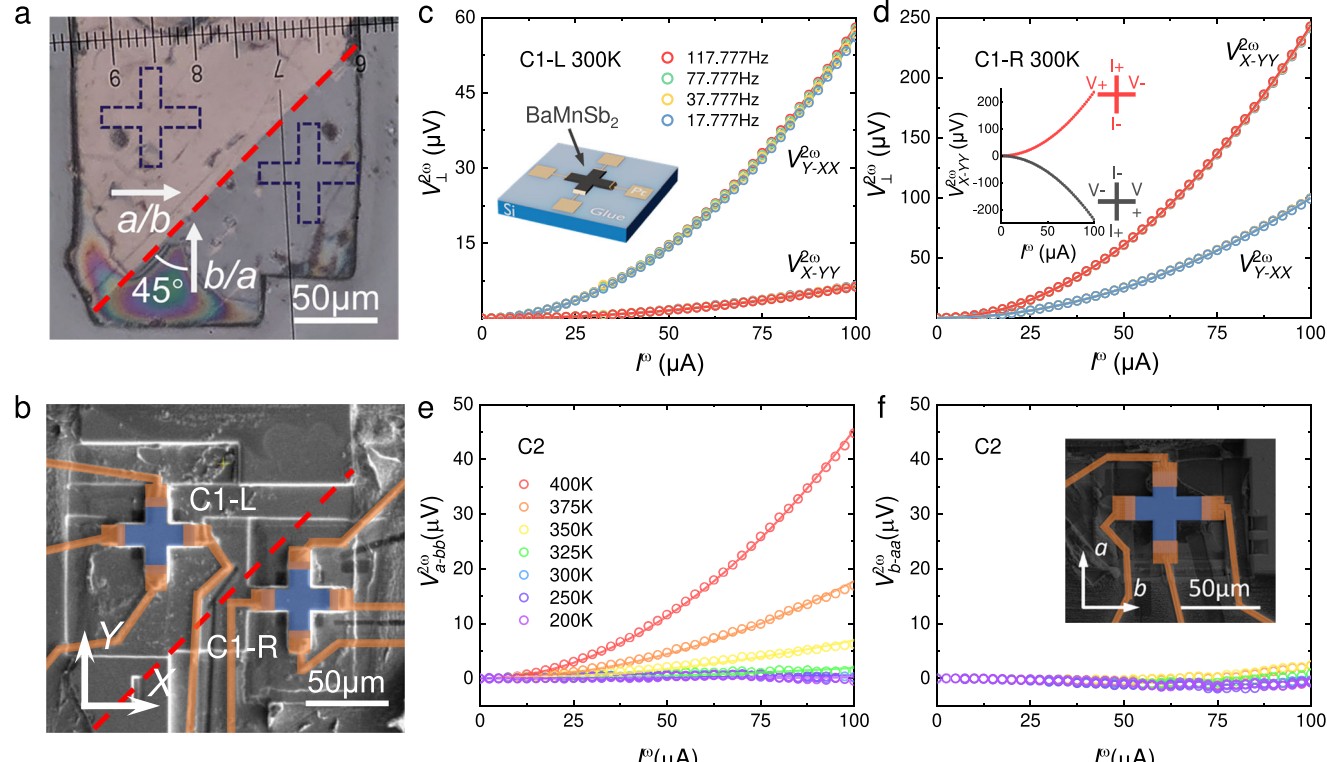

**Fig. 2 | Current direction dependent second-harmonic Hall voltage response measured on cross-like Hall devices of BaMnSb₂. a** Optical image of a BaMnSb$_2$ single-crystalline flake with two major domains. The 90° domain wall is marked with a dashed line. The white arrows indicate the crystallographic axes in both domains. The blue dashed lines show the locations of the two cross-like devices fabricated on this crystal. **b** The SEM image of the cross-like Hall devices carved using FIB on the crystal shown in **a**. Two devices (marked in blue) are located on the two different sides of the domain wall. The deposited Pt contacts and lines are shown in orange. **c, d** The second-harmonic Hall voltage $V_\perp^{2\omega}$ vs. the input ac current $I^\omega$ curves at 300 K for the two devices shown in **b** (C1-L & C1-R). $V_{Y-XX}^{2\omega}$ and $V_{X-YY}^{2\omega}$ refers to $V_\perp^{2\omega}$ measured with $I^\omega$ applied to the $X$- and $Y$-axis, respectively. Both $V_{Y-XX}^{2\omega}$ and $V_{X-YY}^{2\omega}$ scale quadratically with $I^\omega$. $V_{Y-XX}^{2\omega}$ is larger than $V_{X-YY}^{2\omega}$ in C1-L, but smaller than $V_{X-YY}^{2\omega}$ in C1-R. Inset to **c**: schematic of a cross-shaped Hall device. Inset to **d**: $V_\perp^{2\omega}$-$I^\omega$ characteristics of C1-R for the forward current (red curve) and backward current (black curve); the voltage leads' connection is also reversed when the current direction changes from forward to backward. **e, f** $V_\perp^{2\omega}$-$I^\omega$ characteristics at various temperatures for C2, with the current applied to the $b$-axis (**e**) and the $a$-axis (**f**), respectively; while $V_\perp^{2\omega}$ for $I^\omega \parallel b$ ($V_{a-bb}^{2\omega}$) increases significantly above 325 K and quadratically depends on $I^\omega$ (**e**), $V_\perp^{2\omega}$ for $I^\omega \parallel a$ ($V_{b-aa}^{2\omega}$) almost remains zero with increasing temperature (**f**). Inset to **f**: SEM image of C2. Open circles and solid lines in **c**–**e** represent the experiment data and the quadratic fitting, respectively.

find the distributions of the first partial derivative of the Berry curvature, $\partial\mathbf{\Omega}/\partial\mathbf{k}_x$ and $\partial\mathbf{\Omega}/\partial\mathbf{k}_y$, in the momentum space are completely different: $\partial\mathbf{\Omega}/\partial\mathbf{k}_y$ is symmetric between the two Dirac cones, leading to non-zero net BCD, with opposite signs between the conduction and valence bands (Fig. 1f). However, $\partial\mathbf{\Omega}/\partial\mathbf{k}_x$ is antisymmetric along both the $\mathbf{k}_x$ and $\mathbf{k}_y$ directions because of the additional $M_b$ mirror plane. Therefore, the net BCD remains zero when the Fermi surface shifts along $\mathbf{k}_x$. Here, $x$ ($y$) in $\partial\mathbf{\Omega}/\partial\mathbf{k}_x$ ($\partial\mathbf{\Omega}/\partial\mathbf{k}_y$) represents the Fermi surface shift direction in the current-induced nonequilibrium state. As such, a current along the $b$-axis ($y$) can lead to a maximal nonlinear Hall voltage along the $a$-axis, while a current along the $a$-axis ($x$) generates no NLHE.

### Experimental demonstration of the nonlinear Hall effect in BaMnSb₂ microscale devices near room temperature

To verify the predicted NLHE, we fabricated microscale Hall devices of BaMnSb$_2$ via cutting small lamellar crystals using focused ion beam (FIB) and performed nonlinear transport measurements on them using standard lock-in techniques (see Methods). BaMnSb$_2$ crystals used in this study were synthesized using the Sb-flux method (see Methods) and this method enables the growth of thin plate-like crystals with naturally grown flat surfaces (thickness ~3–15 μm), which are critically important for our microscale device fabrication using FIB and NLHE measurements. Most of the samples used in this study have chemical potential close to the conduction band edge (i.e., lightly

electron-doped, see Supplementary Note 1 for more sample information).

We made two types of Hall devices with cross and standard Hall-bar geometries (labeled with C and HB, respectively, hereafter). The inset to Fig. 2c presents a schematic of the cross-shaped Hall devices. Since BaMnSb$_2$ shows 90° twin domains due to its orthorhombic structure and crystals with high domain density could lead the NLHE to be suppressed (see Supplementary Notes 2 & 3), all our Hall devices were made on large domains identified under a polarized microscope. In Fig. 2a, we show a piece of lamellar crystal with two large domains. We made two cross-shaped Hall devices on this crystal with the domain wall lying between them, as shown in Fig. 2a, b. The $X/Y$-axes of both devices are nearly parallel to the crystallographic $a/b$-axis, but the $a/b$-axis orientation of both domains was not determined here though it is known to make an angle of 45° relatives to the domain wall (Fig. 2a). Since theory predicts BaMnSb$_2$ exhibits NLHE only when the driving current is applied to the $b$-axis as illustrated in Fig. 1b and the $a/b$-axis rotates by 90° across the domain wall, we expect to observe NLHE in only one of the devices as the driving current is applied to the $X$- or $Y$-axis of both devices. Our experimental results are overall consistent with this expectation. In both devices, we observed strong second-harmonic Hall voltage ($V_\perp^{2\omega}$) responses under a low frequency (17.777 Hz–117.777 Hz) ac at room temperature; the probed $V_\perp^{2\omega}$ does not show frequency dependence and quadratically depends on $I^\omega$ as expected for the NLHE. However, $V_\perp^{2\omega}$ shows distinct current direction

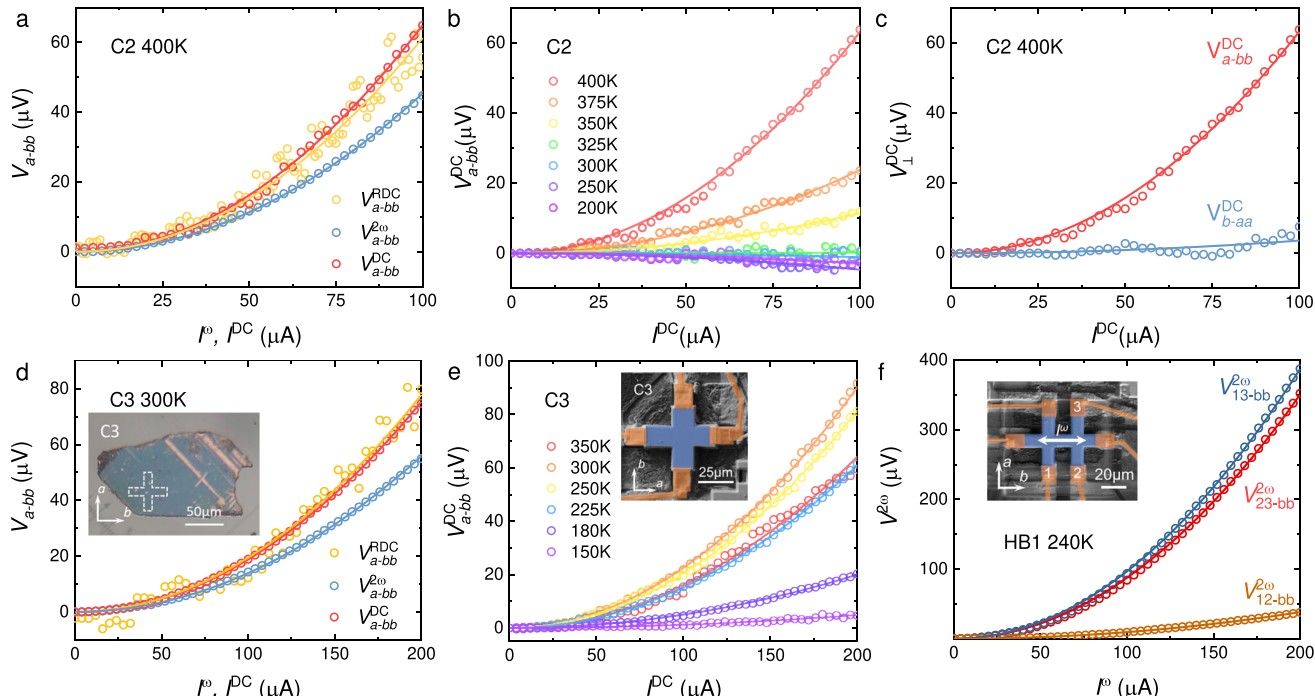

**Fig. 3 | Rectified Hall voltage response driven by ac, nonlinear Hall voltage response driven by dc, and nonlinear Hall angle of BaMnSb$_2$. a** The second-harmonic Hall voltage $V_{a-bb}^{2\omega}$, rectified Hall voltage $V_{a-bb}^{RDC}$, and dc-driven nonlinear Hall voltage $V_{a-bb}^{DC}$ of C2 at 400 K. **b** $V_{a-bb}^{DC}$ at various temperatures for C2. **c** $V^{DC}$ - $I^{DC}$ characteristics at 400 K for C2. $V_{a-bb}^{DC}$ and $V_{b-aa}^{DC}$ refer to $V^{DC}$ measured with the dc $I^{DC}$ applied along the $b$- and $a$-axis, respectively. **d** The second-harmonic Hall voltage $V_{a-bb}^{2\omega}$, rectified Hall voltage $V_{a-bb}^{RDC}$, and dc-driven nonlinear Hall voltage $V_{a-bb}^{DC}$ of C3 at 300 K. Inset to **d**: optical image of the crystal used for fabricating C3. **e** $V_{a-bb}^{DC}$ at various temperatures for C3. Inset to **e**: the SEM image of C3. **f** The second-harmonic voltage $V^{2\omega}$ vs. the input ac current $I^\omega$ characteristics of HB1 at 240 K, measured on three different pairs of voltage leads labeled in the inset. Inset to **f**: the SEM image of HB1. Open circles and solid lines in **a**–**f** represent the experiment data and the quadratic fitting, respectively.

dependence between these two devices. In the left device (labeled by C1-L), $V_\perp^{2\omega}$ for $I^\omega \parallel X$ (expressed by $V_{Y-XX}^{2\omega}$) is ~ 9 times larger than $V_\perp^{2\omega}$ for $I^\omega \parallel Y$ ($V_{X-YY}^{2\omega}$). On the contrary, in the right device (C1-R), the result is the opposite: $V_{X-YY}^{2\omega}$ is 2.4 times larger than $V_{Y-XX}^{2\omega}$. Such a current direction dependence of $V_\perp^{2\omega}$ in both devices is in accordance with the prediction that the NLHE is present only for $I^\omega \parallel b$. The non-zero $V_{X-YY}^{2\omega}$ in C1-L and $V_{Y-XX}^{2\omega}$ in C1-R can be understood as follows: both devices involve some small twin domains, which are not observable on surface imaging; mixed $a$- and $b$-axis oriented domains could lead to the presence of NLHE for both $I^\omega \parallel X$ and $I^\omega \parallel Y$. The distinct current direction dependence of $V_\perp^{2\omega}$ between C1-L and C1-R indicates the $b$-axis of the large domain is approximately oriented along the $x$-axis for C1-L but $y$-axis for C1-R. The different strengths of NLHE between C1-L and C1-R can be attributed to the presence of 90° and 180° domains (see Supplementary Notes 2 & 3). Like the NLHE previously observed in few-layer WTe$_2$[17], the second-harmonic Hall voltage also switches signs when the current direction and the Hall probe connection were reversed simultaneously as illustrated in the inset to Fig. 2d.

To further verify the current direction dependence of $V_\perp^{2\omega}$, we fabricated a cross-shaped Hall device (C2, see the inset to Fig. 2f) on a lamellar crystal whose $a/b$-axis orientation was determined through scanning transmission electron microscopy (STEM) analyses (see Supplementary Note 2). We measured $V_\perp^{2\omega}$ with the current applied along the $b$- and $a$-axis, respectively, (denoted by $V_{a-bb}^{2\omega}$ and $V_{b-aa}^{2\omega}$). As shown in Fig. 2e, f, both $V_{a-bb}^{2\omega}$ and $V_{b-aa}^{2\omega}$ are very small at room temperature (300 K). However, with increasing temperature, $V_{a-bb}^{2\omega}$ shows a significant increase and quadratically scales with $I^\omega$ (Fig. 2e), while $V_{b-aa}^{2\omega}$ remains nearly zero (Fig. 2f). These observations are in good agreement with the expectation of only the $b$-axis current driving NLHE. It is worth pointing out that $V_{a-bb}^{2\omega}$ of C2 at 400 K is much smaller than the room temperature $V_{X-YY}^{2\omega}$ of C1-R at the same driving current. Such a

difference between samples can be attributed to different chemical potentials and domain structures among different samples. This issue as well as the temperature dependence of NLHE will be discussed in detail below.

In C2 device, we have also probed a rectified Hall voltage $V_{a-bb}^{RDC}$, which is $\propto (I^\omega)^2$ and has a magnitude larger than $V_{a-bb}^{2\omega}$, as shown in Fig. 3a. Such a rectification effect is also reproduced in other devices, including C1 (Supplementary Fig. 1), C3 (Fig. 3d), C4 (Supplementary Fig. 2), and HB2 (Supplementary Fig. 3). We further demonstrate such a rectification effect still exists even in the GHz range in C1-R (see Supplementary Note 4 and Supplementary Fig. 4). Besides the ac-driven NLHE, a dc is also expected to drive a nonlinear Hall voltage ($V_{a-bb}^{DC}$) response at zero magnetic field, with $V_{a-bb}^{DC} \propto (I^{DC})^2$ and this was previously demonstrated in the Weyl-Kondo semimetal Ce$_3$Bi$_4$Pd$_3$[34]. In BaMnSb$_2$, we also observed a clear dc-driven nonlinear Hall response with a quadratic $I^{DC}$-$V_{a-bb}^{DC}$ characteristic. Supplementary Fig. 5 plots the $I^{DC}$-$V_\perp^{DC}$ curves measured on C2 & C3 as well as their first-harmonic $I^\omega$-$V_\perp^\omega$ curves. As the lock-in amplifier could screen the second-harmonic signal, the $I^\omega$-$V_\perp^\omega$ curves exhibit a linear behavior as expected, whereas the $I^{DC}$-$V_\perp^{DC}$ curves show nonlinear behavior. This is clearly manifested by the symmetrized transverse voltage data presented in Fig. 3a, d, from which we find $V_{a-bb}^{DC}$ indeed displays a quadratic dependence on $I^{DC}$. Furthermore, we find $V_{a-bb}^{DC}$ is almost the same as $V_{a-bb}^{RDC}$, but ~1.41–1.44 times larger than $V_{a-bb}^{2\omega}$ (Fig. 3a, d). Such a discrepancy between $V_{a-bb}^{DC}$ (= $V_{a-bb}^{RDC}$) and $V_{a-bb}^{2\omega}$ is consistent with what is observed in Ce$_3$Bi$_4$Pd$_3$[34] and close to the expected value of $V_{a-bb}^{RDC}/V_{a-bb}^{2\omega} = \sqrt{2}$ (see Supplementary Note 5). To rule out the possibility of the nonlinear $I$-$V$ curve caused by the heating effect, we also measured $V_{a-bb}^{DC}$ at various temperatures for C2 (Fig. 3b) & C3 (Fig. 3e) and found $V_{a-bb}^{DC}$ vanishes at low temperatures, below 325 K for C2 and 180 K for C3. This is contradictory to the general expectation that the

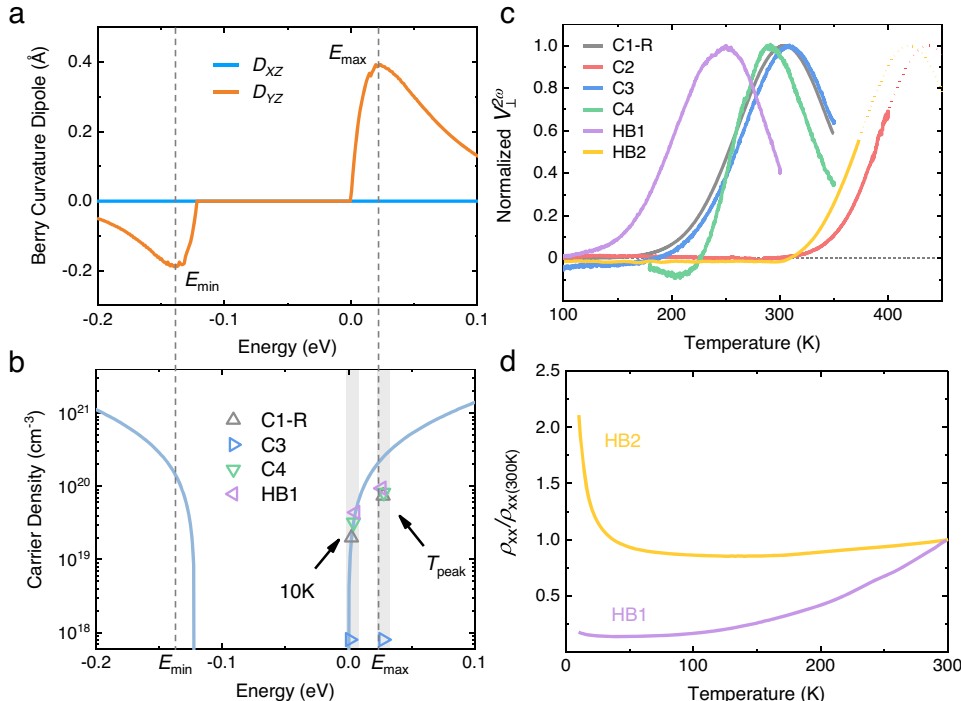

**Fig. 4 | Comparison of the temperature dependence of NLHE with the energy dependence of BCD for BaMnSb₂. a** Calculated energy dependence of BCD. **b** Calculated energy dependence of carrier density and experimental carrier densities measured at 10 K and peak temperatures of the second-harmonic Hall voltage $V_\perp^{2\omega}$ for C1-R, C3, C4 and HB1. **c** Normalized $V_\perp^{2\omega}$ [i.e., $V_\perp^{2\omega}(T)/V_\perp^{2\omega}(T_{peak})$] as a

function of temperature for various samples (C1-R, C2-4 & HB1-2). The curve of HB2 is smoothed. The curves of C2 and HB2 are extrapolated in terms of the peak shape of C1-R. **d** The normalized longitudinal resistivity as a function of temperature for HB1 and HB2.

heating effect becomes severe at low temperatures. However, this can be well understood in terms of the temperature dependence of intrinsic NLHE of BaMnSb₂ as to be discussed below. Like $V_\perp^{2\omega}$, $V^{DC}$ also exhibits a distinct current direction dependence in C2, namely, $V_{a-bb}^{DC} \propto (I^{DC})^2$ for $I^{DC} \parallel b$, while $V_{b-aa}^{DC}$ is nearly zero for $I^{DC} \parallel a$, as shown in Fig. 3c [Note that C2 & C3 are our recently prepared devices and we observed dc nonlinear transport first on these two samples. When we went back to verify dc nonlinear transport in other samples (C1, C4, HB1 & HB2) prepared in the early stage of this research (6–12 months old), we found those samples already degraded (see Supplementary Note 6).

Another distinct hallmark of the NLHE is the nonlinear Hall angle $\theta_{NLHE}$ of 90°[13]. $\theta_{NLHE}$ is defined as $\arctan(V_\perp^{2\omega}/V_\parallel^{2\omega})$ where $V_\parallel^{2\omega}$ represents the longitudinal voltage at the frequency of 2ω. In an ideal Hall-bar device that shows an intrinsic NLHE and does not involve mixing of the voltage components along the longitudinal and transverse directions, a longitudinal ac, when applied along the BCD direction, should drive a second-harmonic Hall voltage response only along the transverse direction, but not along the longitudinal direction, i.e., $V_\perp^{2\omega} \neq 0$ and $V_\parallel^{2\omega} = 0$, thus $\theta_{NLHE} = 90°$. We have demonstrated that $\theta_{NLHE}$ is indeed close to 90° for the NLHE of BaMnSb₂. In general, measuring $\theta_{NLHE}$ in bulk single crystals is challenging, since the unavoidable misalignments of the voltage test leads can lead the longitudinal voltage to involve a Hall voltage component, thus engendering the deviation of $\theta_{NLHE}$ from 90°. To demonstrate $\theta_{NLHE}$ of 90°, we fabricated a standard Hall-Bar device HB1 (inset to Fig. 3f) on a crystal with determined $a$- and $b$-axis orientations. Figure 3f shows the second-harmonic voltages measured on three different pairs of voltage leads for HB1, e.g., $V_{12-bb}^{2\omega}$, $V_{23-bb}^{2\omega}$, and $V_{13-bb}^{2\omega}$. Here, the subscript numbers (1, 2, & 3) represent the Hall voltage leads shown in the inset to Fig. 3f (Note that the lead without a label in this device was broken during cooling down). All these measured second-harmonic voltages quadratically depend on the driving current. The transverse voltage $V_{23-bb}^{2\omega}$ and diagonal voltage $V_{13-bb}^{2\omega}$ have similar

magnitudes, and they are much larger than the longitudinal voltage $V_{12-bb}^{2\omega}$. These data not only demonstrate the nonlinear Hall voltage's dominance over the longitudinal response but also show $\theta_{NLHE} = \arctan(V_{23-bb}^{2\omega}/V_{12-bb}^{2\omega})$ is about 84°, which is very close to the expected 90° and independent of current. The $\theta_{NLHE}$ measured in the other standard Hall-bar device (HB2) is ~52° (see Supplementary Fig. 3); its large deviation from 90° is likely caused by misalignments of the voltage test leads and the inhomogeneity of the sample. The demonstration of $\theta_{NLHE} \approx 90°$ in HB1 clearly indicates that the second-harmonic Hall voltage response observed in our experiments arises from the NLHE. These data also exclude the possibility that our observed nonlinear Hall response in BaMnSb₂ is associated with extrinsic effects such as the contact junction effect, asymmetric sample shape, and thermoelectric effect[13] (see more discussions in Supplementary Note 7).

As noted above, NLHE could arise either from BCD (intrinsic origin) or from spin-dependent scattering (extrinsic origin). From the comparison of the temperature dependence of second-harmonic Hall voltage $V_\perp^{2\omega}$ with the calculated energy dependence of BCD, we find the NLHE of BaMnSb₂ has an intrinsic origin. Figure 4a presents our calculated BCD near the bandgap for BaMnSb₂. In the valance band, the BCD is negative. As the energy goes up, the BCD first decreases to a minimum value of −0.20 Å at $E_{min}$ and then increases to zero at the band top. On the other hand, the BCD in the conduction band is positive. The BCD rises sharply with the energy from the band bottom and then starts to decrease monotonically after reaching a maximum of 0.39 Å at $E_{max}$ (~23 meV). Inside the bandgap, the BCD remains zero. These calculation results suggest that if our samples have $E_F$ close to the conduction band edge (i.e., lightly electron-doped), $V_\perp^{2\omega}$ would be expected to show a nonmonotonic temperature dependence and reaches a maximum near room temperature. This is because the electrons at $E_F$ under this circumstance can be thermally excited to $E_{max}$ at ~300 K where BCD is maximized (Note that $k_B T = 26$ meV for

$T = 300$ K). This is exactly what we observed in experiments. Several BaMnSb$_2$ samples used in our study, including C1 (Fig. 2b–d), C3 (Fig. 3d, e), C4 (Supplementary Fig. 2), and HB1 (Fig. 3f), are indeed lightly electron-doped as evidenced by the conventional Hall measurements (Supplementary Note 1). From the comparison of the calculated and measured carrier density (Fig. 4b), we find the $E_F$ for most of these samples is slightly above the conduction band bottom. Their temperature dependencies of $V_\perp^{2\omega}$ do show maxima near room temperature, as shown in Fig. 4c where $V_\perp^{2\omega}$ is normalized to its peak value for comparison. The carrier density measured at the peak temperatures is also close to the calculated values except for C3. The differences in peak temperatures among these samples can be attributed to their slight differences in Fermi energy due to non-stoichiometric compositions[40]. Importantly, despite small differences in $E_F$ among these samples, their $V_\perp^{2\omega}$ peaks share similar shapes except for C4, suggestive of their common intrinsic origin. It is worth noting while C3 displays a $V_\perp^{2\omega}$ peak similar to those of C1, C4, and HB1, its carrier density is one order of magnitude smaller than those of C1, C4, and HB1, implying that this sample is inhomogeneous, with $E_F$ being inside the Dirac gap for the major fraction of this sample.

Sample C4 shows not only a slightly different peak profile from other samples, but also a clear sign change near 220 K. This can be possibly interpreted by 180° domains and slight chemical inhomogeneity (see Supplementary Note 3 for detailed discussions). Additionally, another noteworthy result is that samples C2 and HB2 do not show maximal values in $V_\perp^{2\omega}$ even as the temperature is increased to 400 K(C2) and 373 K(HB2). This can be ascribed to the fact that $E_F$ of C2 and HB2 is inside the Dirac gap, which is evidenced by their lower carrier densities (Supplementary Table 1) and insulating-like behavior below 100 K in the temperature dependence of in-plane resistivity measured in HB2, sharply contrasted with the metallic transport for HB1 in the whole measured temperature range (Fig. 4d). Its lower $E_F$ leads the thermal excitation energy of electrons cannot reach $E_{max}$ even at 400 K. Based on the temperature dependence of $V_\perp^{2\omega}$ of sample C1-R, we extrapolated the $V_\perp^{2\omega}$ curves of HB2 & C2, from which their peaks are estimated to be in the 420–440 K range. The consistency of the observed $V_\perp^{2\omega}$ peaks near room temperature with the calculated energy dependence of BCD provide strong evidence that the NLHE in BaMnSb$_2$ is intrinsic. Its intrinsic origin is further corroborated by the nonmonotonic dependence of the NLHE's strength [defined as $E_\perp^{2\omega}/(E_\parallel^\omega)^2$ where $E_\perp^{2\omega}$ and $E_\parallel^\omega$ represents the values of transverse and longitudinal electric fields, respectively] on the longitudinal conductivity $\sigma_{xx}$ (see Supplementary Note 8). In general, $E_\perp^{2\omega}/(E_\parallel^\omega)^2$ is expected to be linearly dependent on $(\sigma_{xx})^2$ if the NLHE is dominated by skew scattering[23]. Our observation of a peak in the $\sigma_{xx}$ dependence of $E_\perp^{2\omega}/(E_\parallel^\omega)^2$ (Supplementary Fig. 12) clearly excludes the skew scattering mechanism as a main contribution, but can find interpretation from the energy dependence of BCD in Fig. 4a.

Although the peak of $V_\perp^{2\omega}$ near room temperature in BaMnSb$_2$ originates from the nonmonotonic energy dependence of BCD, it does not appear at the temperature where the BCD is maximized. This can be understood as follows. According to the theory[13], the BCD is proportional to $E_\perp^{2\omega}/(E_\parallel^\omega)^2$. Supplementary Fig. 13 plots the temperature dependences of $E_\perp^{2\omega}/(E_\parallel^\omega)^2$ and $V_\perp^{2\omega}$ together for sample HB1. While $V_\perp^{2\omega}$ shows a peak at 250 K, $E_\perp^{2\omega}/(E_\parallel^\omega)^2$ peaks at 206 K. This is because that $V_\perp^{2\omega}$ depends not only on BCD but also on the linear conductivity $\sigma^{(1)}$; that is, $V_\perp^{2\omega} \propto \mathbf{D}/[\sigma^{(1)}]^2$ (see Supplementary Note 9 for more details). Given that $\sigma^{(1)}$ monotonically decreases with increasing temperature (also shown in Supplementary Fig. 13), the peak of $V_\perp^{2\omega}$ shifts to a higher temperature.

Furthermore, we have also estimated the maximal 2D and 3D nonlinear conductivities of sample HB1 using $\sigma^{(2)} = -\sigma^{(1)}E_\perp^{2\omega}/(E_\parallel^\omega)^2$[25]. As shown in Supplementary Note 9 and Supplementary Fig. 13, the maximal 2D NLH conductivity of HB1 is 26.4 nm·S/V, which is two to three orders of magnitude smaller than that of graphene moiré superlattices[25] but much larger than those of the bilayer (0.9 nm·S m$^{-1}$) and five-layer WTe$_2$ (3 pm·S m$^{-1}$)[13,17,25]. Its maximal 3D nonlinear Hall conductivity near room temperature is about 22 S V$^{-1}$, larger than that of TaIrTe$_4$[14]. As indicated above, larger nonlinear Hall conductivity is critical for applications.

The inversion symmetry breaking in BaMnSb$_2$ plays a critical role in generating its room-temperature NLHE. This is corroborated by the absence of NLHE in SrMnSb$_2$, which also hosts a massive Dirac fermion state, but possesses a centrosymmetric orthorhombic structure[41,42] (see Supplementary Note 10). Finally, it should be pointed out that the NLHE of BaMnSb$_2$ is significantly impacted by 90° and 180° domains, whose characteristics are revealed by STEM studies (see Supplementary Note 2). Both domain types suppress the NLHE (see Supplementary Note 3). Samples with high density 180° domains would have strongly suppressed NLHE due to the cancellation of the nonlinear Hall voltage between domains. The sample dependence of NLHE observed in our experiments can be attributed to the variations of the domain structures and $E_F$ among different samples. If BaMnSb$_2$ could be grown to thin films without domain walls, its NLHE's strength would be expected to increase by one order of magnitude (Supplementary Note 3), which might open a door to technological applications of NLHE.

## Wireless frequency doubling based on the observed nonlinear Hall effect

Finally, we demonstrate room-temperature wireless microwave detection and frequency doubling based on the observed NLHE. Figure 5a shows the experimental setup. A signal generator was used to generate a radio frequency (RF) signal with a frequency of 300 MHz.

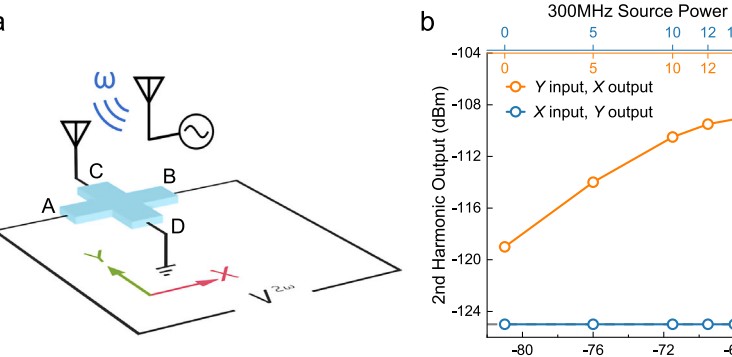

**Fig. 5 | Room-temperature wireless microwave frequency doubling based on the NLHE. a** Schematic of the experiment setup. A pair of antennas are used to transmit and receive wireless signals. Panel **b** shows the measured second-harmonic output power vs. the measured fundamental (first-harmonic) power. The output power from the signal generator is marked on the top axis. The gray dashed line indicates the noise floor.

A pair of antennas, respectively, connected to the signal generator and cross-like Hall device C1-R (Fig. 2b) was employed to transmit and receive wireless RF signals. When the antenna was connected to the terminals along the $y$-axis of device C1-R, the signal analyzer connected to the terminals along the $x$-axis probed a striking second-harmonic signal. Figure 5b shows the measured second-harmonic power $P_{AB-CD}^{2\omega}$ vs. the fundamental power measured between the A & B terminals; $P_{AB-CD}^{2\omega}$ increases with the fundamental power. However, when the antenna was switched to the A & B terminal, the second-harmonic power probed between the C & D terminals $P_{CD-AB}^{2\omega}$ was at the noise level or 10 times smaller than $P_{AB-CD}^{2\omega}$. These results not only verify the current direction dependence of the NLHE discussed above, but also demonstrate such an NLHE-based device indeed acts as an RF frequency doubler.

## Methods

### Crystal growth

Single-crystalline micro flakes of BaMnSb$_2$ were synthesized using an Sb-flux method. High-purity Ba pieces, Mn, and Sb powders were mixed in the molar ratio of 1:1:3 or 1:1:4, placed in alumina crucibles, and sealed in evacuated quartz tubes. The tubes were heated up to 1050 °C in a furnace for one day and then cooled down to 1000 °C, followed by further cooling down to 700 °C at a rate of 3 °C/h. After the tubes were centrifuged to remove excess Sb at 700 °C, single-crystalline flakes with thicknesses around 10 μm were obtained, along with some larger plate-like single crystals.

### Device fabrication

Microscale devices in the cross and the standard Hall-bar geometry were fabricated by focused ion beam (FIB) cutting using FEI Helios NanoLab 660 and FEI Scios 2 dual beam SEM. The platinum electric contacts were in situ made inside the SEM chamber. To make sure the contact is good throughout the whole thickness, the contact areas were first fabricated into wedge shapes before the Pt deposition. After the cutting and deposition processes, a final cleaning step was conducted to remove the redeposition materials.

### Electrical transport measurements

The electrical transport measurements were conducted using a Quantum Design Physical Property Measurement System (PPMS). The driving current was generated by a Keithley 6221 precision AC/DC current source or a Stanford Research SR860 lock-in amplifier. The ac and dc voltages were measured by a Stanford Research SR860 lock-in amplifier and a Keithley 2182A nanovoltmeter, respectively. The rectified Hall voltage is usually hard to measure. The reason is that in the low-frequency range, the rectified Hall voltage generated by the NLHE is mixed with the input signal $V^{\omega}$, as well as the second-harmonic signal $V_{\perp}^{2\omega}$ resulting from the NLHE and the nanovoltmeter used to measure the rectified voltage cannot filter off $V^{\omega}$ and $V_{\perp}^{2\omega}$. As such, the measured rectification voltage would be very noisy. We found that increasing the frequency of the driving current can alleviate this problem. The rectification voltages presented in this paper were measured with an input current of 117.777 Hz or 137.777 Hz for all the devices.

### STEM analyses

The STEM samples were prepared using a FIB system. Two cross-sectional lamellas were lifted out from the crystals, which were used for fabricating HB1 and HB2. HAADF-STEM images are taken with the Thermo Fisher Titan3 S/TEM equipped with a spherical aberration corrector. The atomic displacement analysis[43] was performed to quantitatively analyze the local domain structures.

### Calculation method

The electronic structures, Berry curvature $\boldsymbol{\Omega}(\mathbf{k})$, and Berry curvature dipole $d\boldsymbol{\Omega}/d\mathbf{k}$ are calculated with an eight-band tight-binding model for an Sb layer in the conventional unit cell. The detailed model was presented previously[37]. While the model is kept the same, some parameters are optimized to reproduce the density-functional theory[44] (DFT) band structure more accurately at the Dirac point on the X-M line in the Brillouin zone. The set of all parameters employed in this work are: $\tilde{m}_0 = 0$, $\tilde{m}_1 = 0.48$ eV, $t_0 = 0.97$ eV, $t_1 = 1$ eV, $t_2 = -0.1$ eV, $t_3 = 0.1$ eV, $t_4 = -0.04$ eV, $t_5 = 0.145$ eV, $t_6 = -0.04$ eV, and $\lambda_0 = 0.25$ eV. The meaning of these parameters can be found in the previous results[37]. Notice that only the out-of-plane component $\boldsymbol{\Omega}_{xy}$ of the Berry curvature is meaningful since the model for the two-dimensional Sb layer is employed here. The BCD components of $d\boldsymbol{\Omega}/d\mathbf{k}_x, d\boldsymbol{\Omega}/d\mathbf{k}_y$ are evaluated at different Fermi energies.

### Radio frequency measurement

A Keysight MXG Vector signal generator was used to generate RF signals. A pair of TP-link antennas were used to transmit and receive wireless signals. The receiving antenna connects to one arm of the cross-like device made of BaMnSb$_2$ while the opposite arm was grounded. The signals from the two terminals in the other direction were measured by an Agilent CXA signal Analyzer. In the rectification experiment, the RF signal generator was directly connected to the cross-like device, while the dc signal was measured by a voltmeter.

## Data availability

The authors declare that all the data that support the findings of this study are available within the paper and Supplementary Information. Additional relevant data are available from the corresponding authors upon reasonable request. Source data are provided with this paper.

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

## Acknowledgements
Z.M., N.A., L. Min, V.G., and L. Miao acknowledge the support from NSF through the Materials Research Science and Engineering Center DMR 2011839 (2020–2026). Partial financial support for Z.M. and L. Min on this project is provided by the US National Science Foundation under grant DMR 2211327. B.Y. and H.T. acknowledge funding from the European Research Council (ERC) under the European Union's Horizon 2020 research and innovation programme (ERC Consolidator Grant "NonlinearTopo", No. 815869). S.H.L. acknowledges the support provided by the National Science Foundation through the Penn State 2D Crystal Consortium-Materials Innovation Platform (2DCC-MIP) under NSF cooperative agreement DMR-2039351. N.A. and L. Mia acknowledge the Air Force Office of Scientific Research (AFOSR) program FA9550-18-1-0277 as well as GAME MURI, 10059059-PENN for support. We appreciate the support and resources from Materials Characterization Lab (MCL) at Penn State.

## Author contributions
Z.M. and L. Min conceived the project. The crystal growth and transport measurements were carried out and analyzed by L. Min, R.Z., S.H.L., and Z.M. The device fabrications were performed by L. Min, V.G., and Z.M. The electronic structures, Berry curvature, and Berry curvature dipole were calculated by H.T., C.X.L., and B.Y. Z.X. performed the wireless microwave detection experiment. The TEM study was performed by L. Miao and N.A. The paper was written by L. Min and Z.M. with inputs from all authors.

## Competing interests
The authors declare no competing interests.
