## [Peer Review File · Nature Communications]

Strong Room-Temperature Bulk Nonlinear Hall Effect in a Spin-Valley Locked Dirac MaterialREVIEWER COMMENTS

Reviewer #1 (Remarks to the Author):

The nonlinear Hall effect has attracted great attention recently due to its deep connection with the Berry curvature and the potential applications in microwave generation and detection at rare frequency ranges. In this work, the authors investigated the nonlinear Hall effect in BaMnSb₂. They observed both ac and dc nonlinear Hall signal at the room temperature. In the previous research, TaIrTe₄ is the only material that exhibits the nonlinear Hall effect at room temperature, but due to the surface contribution. In contrast, the authors show that the nonlinear Hall effect in BaMnSb₂ is originated from the bulk Berry curvature dipole. Because 2D surface vs. 3D bulk is like 0 to infinity, the bulk contribution in BaMnSb₂ is much greater. Also, this work demonstrates as far as I know the first experiment on the microwave generation (probably the first on any of the applications) using the nonlinear Hall effect.

Overall, I find the results interesting, novel, and important. The theoretical and experimental results seem consistent and BaMnSb₂ is a better material than TaIrTe₄ from the application point of view. Thus, I can recommend the publication of this manuscript after the following question is addressed.

1. The author should also provide the data about the nonlinear conductivity besides the nonlinear voltage. Is it better to compare the maximum nonlinear conductivity of BaMnSb₂ and TaIrTe₄ at the room temperature, like Table 1 in a recent work [He et al. Nature Nanotechnology 17. 378-383 (2022)].
2. If I got it right, in terms of the nonlinear conductivity, the double-frequency response should be in principle the same as the dc response, thus no $\pi/2$ factor difference if the data is not presented in terms of the voltage.

Reviewer #2 (Remarks to the Author):

Min et al. reports on a nonlinear hall effect in BaMnSb₂ at room temperature. Nonlinear Hall effect has attracted significant efforts over the past few years because it detects important geometric properties of quantum wavefunction in topological materials. Moreover, this effect is highly relevant to technology applications such as wireless energy harvesting. So it is desirable to achieve high nonlinear power transduction at room temperatures. So far only TaIrTe₄ has been reported to exhibit sizable room temperature nonlinear Hall response (Nature Nanotechnology 16, 421–425 (2021)). The authors claimed that BaMnSb₂ is the second example that exhibits room temperature nonlinear Hall conductivity. Moreover, there is a noticeable distinction between TaIrTe₄ and BaMnSb₂ in that the effect in TaIrTe₄ has to be a thin-film effect since the bulk has a higher symmetry that prevents the development of the nonlinear Hall in the plane. I found the temperature dependence is particularly interesting. For previous works, the nonlinear hall signal always gets smaller as the temperature increases. However, here the nonlinear Hall signal is actually negligible below room temperature and increases significantly above. This phenomenon is interpreted as thermal excitation of electronic states to Berry curvature hot spot.

The experiment is systematic and solid. But I am a little skeptical about the interpretation of the temperature dependence. The strength of Berry curvature dipole should be directly proportional to the nonlinear-Hall conductivity, which should roughly scale with $V_{2w}/(I^2 R^3)$. If R changes significantly with temperature, the temperature dependence of V_{2w} may not be a good indication of the Berry curvature.

In addition, for the DC current-DC voltage measurement, I wonder what is situation on the opposite

side (i.e., $I_{dc} < 0$).

Otherwise, I found the paper is interesting, well-written and with sufficient references. I recommend its publication if the authors address my concerns above.

Reviewer #3 (Remarks to the Author):

Min et al. reported observation of the nonlinear Hall effect in a candidate Dirac material BaMnSb₂. Second harmonic and rectified Hall voltages quadratic to the applied current were observed. The effect is found to be stronger when current is applied along the crystal b-axis, but much weaker when the current is along the crystal a-axis. These are solid evidence for the nonlinear Hall effect. However, the manuscript may not be suitable for publication in Nature Communications in its current form for the following reasons:

1, Nonlinear Hall effect has been reported in bulk materials, such as Weyl semimetals (ref. 33, 34). The observed nonlinear Hall effect here is also not that different from the strained transition metal dichalcogenides in Ref. 29, which are also massive Dirac materials. Moreover, the effect has been observed at high temperatures (ref. 14). These studies compromise the novelty of this work.

2, The authors claim that the observed nonlinear Hall effect is an intrinsic Berry curvature effect based on the non-monotonic temperature dependence. They claim the material is slightly electron-doped at low temperatures, and the carrier density increases with temperature due to thermal activations. However, this claim is questionable. First, the authors only showed carrier densities at temperatures of 10K and near the nonlinear Hall maximum. How does the carrier density change at higher temperatures? Is the carrier density change at higher temperatures consistent with the rapid reduction of the nonlinear Hall voltage? Second, the nonlinear Hall voltage switches sign as sample temperature varies, most evidently in sample C4, at $T \sim 200\text{K}$. Such sign reversal in NLHE is not expected unless carrier type changes or band structure renormalizes.

Response Letter

We appreciate all three referees for reviewing our manuscript. We are grateful for their comments and constructive suggestions, which have been very helpful in improving our manuscript. In the following, we provide a point-to-point response (shown in blue) to all comments raised by the referees.

Response to Reviewer #1

Reviewer #1 (Remarks to the Author):

The nonlinear Hall effect has attracted great attention recently due to its deep connection with the Berry curvature and the potential applications in microwave generation and detection at rare frequency ranges. In this work, the authors investigated the nonlinear Hall effect in BaMnSb₂. They observed both ac and dc nonlinear Hall signal at the room temperature. In the previous research, TaIrTe₄ is the only material that exhibits the nonlinear Hall effect at room temperature, but due to the surface contribution. In contrast, the authors show that the nonlinear Hall effect in BaMnSb₂ is originated from the bulk Berry curvature dipole. Because 2D surface vs. 3D bulk is like 0 to infinity, the bulk contribution in BaMnSb₂ is much greater. Also, this work demonstrates as far as I know the first experiment on the microwave generation (probably the first on any of the applications) using the nonlinear Hall effect.

Overall, I find the results interesting, novel, and important. The theoretical and experimental results seem consistent and BaMnSb₂ is a better material than TaIrTe₄ from the application point of view. Thus, I can recommend the publication of this manuscript after the following question is addressed.

Response: We thank the referee for the concise summary of our work and the positive assessment of our work. We also appreciate the referee's questions, which have been extremely helpful in improving the manuscript.

1. The author should also provide the data about the nonlinear conductivity besides the nonlinear voltage. Is it better to compare the maximum nonlinear conductivity of BaMnSb₂ and TaIrTe₄ at the room temperature, like Table 1 in a recent work [He et al. Nature Nanotechnology 17. 378-383 (2022)].

Response: We thank the referee for bringing the recent work by He et al. [Nature Nanotechnology 17. 378-383 (2022)] to our attention. According to that paper, the nonlinear Hall (NLH) conductivity is described by $\sigma^{(2)} = -\sigma^{(1)}V_{\perp}^{(2)}L^2/(V_{\parallel}^2W)$, where $\sigma^{(1)}$ is the 2D linear conductivity, $V_{\perp}^{(2)}$ is the transverse second-order voltage, V_{\parallel} is the voltage of frequency ω along the longitudinal direction, and L and W are the length and the width of the channels, respectively. Therefore, we can further convert it into $\sigma^{(2)} = -\sigma^{(1)}E_{\perp}^{2\omega}/(E_{\parallel}^{\omega})^2$ where $E_{\perp}^{2\omega}$ and E_{\parallel}^{ω} represents transverse and longitudinal electric fields respectively. From the data of $E_{\perp}^{2\omega}/(E_{\parallel}^{\omega})^2$ and the 3D linear conductivity shown in Supplementary Fig. S12, we have calculated the temperature dependence of 3D and 2D NLH conductivity, as shown in Fig. R1 (attached below). In the 2D NLH conductivity calculation, the normalized linear conductivity, $\sigma^{(1)}/Sb$ layer is used. Fig. R1 shows the maximal 2D NLH conductivity value is 26.4 nm·S/V, which is two to three orders of magnitude smaller than that of graphene moiré superlattices reported in the above reference paper, but much larger than those of the bilayer (0.9 nm·S/m) and five-layer WTe₂ (3 pm·S/m). As for the NLHE of TaIrTe₄, we estimated only its 3D NLH conductivity in the low temperature limit (at 2K) since ref. 14

reported only low temperature linear conductivity data for those samples showing NLHE (Supplementary Fig. S7 in ref. 14). Our estimated 3D NLH conductivity for the 16-53 nm TaIrTe₄ samples reported in ref. 14 is ~ 0.01 - 0.04 S/V. Given that TaIrTe₄ has comparable 2nd harmonic Hall voltage responses between 2 K and 300K and its linear conductivity at room temperature is smaller than that of 2K, its 3D NLH conductivity at 300K should be smaller than 0.01 - 0.04 S/V. In contrast, the maximal 3D NLH conductivity of BaMnSb₂ near room temperature is ~ 22.5 S/V(see Fig. R1), three orders of magnitude larger than that of TaIrTe₄. We have added this comparison to the revised manuscript on page 6 and Fig. R1 to Supplementary Fig. S13. We have also cited the recent work by He et al. [Nature Nanotechnology 17. 378-383 (2022)] in the main text. We really appreciate the referee's mention of this work, which motivated us to perform further analyses and better understand the essential differences of the room temperature NLHE between BaMnSb₂ and TaIrTe₄.

Fig. R1: Temperature-dependent 3D and 2D NLH conductivity of BaMnSb₂ measured on sample HB1

2. If I got it right, in terms of the nonlinear conductivity, the double-frequency response should be in principle the same as the dc response, thus no $\pi/2$ factor difference if the data is not presented in terms of the voltage.

Response: We thank the reviewer for raising this thoughtful question. According to the theory (ref. 11), the nonlinear Hall current can be expressed as $j^{NLH} \propto \sin^2(\omega t) = \frac{1}{2} - \frac{1}{2} \cos(2\omega t)$. Here the first and second terms represent the DC rectified and 2nd harmonic Hall current respectively. This indicates that the rectified Hall voltage V_{\perp}^{RDC} should be the same as the peak value of the 2nd harmonic voltage $V_{\perp}^{2\omega, peak}$. However, the 2nd harmonic voltage measured by a lock-in amplifier is the root-mean-square (RMS) voltage $V_{\perp}^{2\omega}$ which is equal to $V_{\perp}^{2\omega, peak} / \sqrt{2} = V_{\perp}^{RDC} / \sqrt{2}$. This explains why our measured V_{\perp}^{RDC} is about 1.4 times larger than $V_{\perp}^{2\omega}$. The previously reported ratio of $V_{\perp}^{RDC} / V_{\perp}^{2\omega} = 1.41$ for Ce₃Bi₄Pd₃ in ref. 34 should be of the same origin. In the Supplementary Note 5 of the original manuscript, we made a mistake and mistook the 2nd harmonic RMS voltage as the time average of the $V^{2\omega}$, so we derived a wrong equation of $V_{\perp}^{RDC} / V_{\perp}^{2\omega} = \pi/2$. We have corrected this mistake in the Supplementary Note 5 of the current manuscript.

As pointed out above, the nonlinear Hall conductivity can be expressed as $\sigma_{yxx}^{(2)} = -\sigma^{(1)}V_y^{(2)}L^2/(V_0^2W)$, which is still dependent on the transverse Hall voltage term $V_y^{(2)}$. Consequently, the difference between V_{\perp}^{RDC} and $V_{\perp}^{2\omega}$ is still manifested in the nonlinear Hall conductivity. We have made this clear in the revised manuscript.

We hope we have satisfactorily addressed all the issues raised by the reviewer.

Response to Reviewer #2

Reviewer #2 (Remarks to the Author):

Min et al. reports on a nonlinear hall effect in BaMnSb2 at room temperature. Nonlinear Hall effect has attracted significant efforts over the past few years because it detects important geometric properties of quantum wavefunction in topological materials. Moreover, this effect is highly relevant to technology applications such as wireless energy harvesting. So it is desirable to achieve high nonlinear power transduction at room temperatures. So far only TaIrTe4 has been reported to exhibit sizable room temperature nonlinear Hall response (Nature Nanotechnology 16, 421–425 (2021)). The authors claimed that BaMnSb2 is the second example that exhibits room temperature nonlinear Hall conductivity. Moreover, there is a noticeable distinction between TaIrTe4 and BaMnSb2 in that the effect in TaIrTe4 has to be a thin-film effect since the bulk has a higher symmetry that prevents the development of the nonlinear Hall in the plane. I found the temperature dependence is particularly interesting. For previous works, the nonlinear hall signal always gets smaller as the temperature increases. However, here the nonlinear Hall signal is actually negligible below room temperature and increases significantly above. This phenomenon is interpreted as thermal excitation of electronic states to Berry curvature hot spot.

The experiment is systematic and solid. But I am a little skeptical about the interpretation of the temperature dependence. The strength of Berry curvature dipole should be directly proportional to the nonlinear-Hall conductivity, which should roughly scale with $V_{2w}/(l^2 R^3)$. If R changes significantly with temperature, the temperature dependence of V_{2w} may not be a good indication of the Berry curvature.

Response: We thank the referee for taking time to review our manuscript. We also appreciate the insightful comments made by the referee, which have been very helpful in improving the manuscript.

Previous work has shown the nonlinear Hall conductivity $\sigma^{(2)} = -\sigma^{(1)}V_{\perp}^{(2)}L^2/(V_{\parallel}^2W)$ (He et al. Nat. Nanotech. 17, 378 (22)), where $\sigma^{(1)}$ is the 2D linear conductivity, $V_{\perp}^{(2)}$ is the transverse second-order Hall voltage, V_{\parallel} is the voltage of frequency ω along the longitudinal direction, and L and W are the length and the width of the channels, respectively. We can further convert it into $\sigma^{(2)} = -\sigma^{(1)}E_{\perp}^{2\omega}/(E_{\parallel}^{\omega})^2$ (1), where $E_{\perp}^{2\omega}$ and E_{\parallel}^{ω} represents transverse and longitudinal electric fields respectively and $E_{\perp}^{2\omega}/(E_{\parallel}^{\omega})^2$ is proportional to the Berry curvature dipole \mathbf{D} . \mathbf{D} can be extracted via the equation $\mathbf{D} = \frac{2\hbar^2\sigma^{(1)^3}V_{\perp}^{(2)}W}{e^3\tau(J_{\parallel}^{\omega})^2}$ (2) (ref. 13)

where τ is the scattering time which is proportional to $\sigma^{(1)}$. In Fig. R2, we plot the temperature dependences of the normalized $V_{\perp}^{2\omega}$, $E_{\perp}^{2\omega}/(E_{\parallel}^{\omega})^2$, $\sigma^{(1)}$, and $\sigma^{(2)}$ together for sample HB1. From 100K to 300K where we observed the nonlinear Hall effect, $\sigma^{(1)}$ shows a monotonic decrease with increasing temperature, while $V_{\perp}^{2\omega}$, $E_{\perp}^{2\omega}/(E_{\parallel}^{\omega})^2$ and $\sigma^{(2)}$ show maxima. Such a contrast clearly shows that the variation of linear conductivity (or the resistance) with temperature is not the reason why we observed a peak in the nonlinear Hall response. Instead, it is the temperature dependence of \mathbf{D} ($\propto E_{\perp}^{2\omega}/(E_{\parallel}^{\omega})^2$) that results in a peak in the temperature dependence of nonlinear Hall voltage response. It is worth noting that the peak temperatures of $V_{\perp}^{2\omega}$, $E_{\perp}^{2\omega}/(E_{\parallel}^{\omega})^2$, and $\sigma^{(2)}$ are different: while $E_{\perp}^{2\omega}/(E_{\parallel}^{\omega})^2$ peaks at 206K, $V_{\perp}^{2\omega}$ and $\sigma^{(2)}$ show peaks at 250K and 193K respectively. This is because that $V_{\perp}^{2\omega}$ and $\sigma^{(2)}$ depend not only on \mathbf{D} but also on $\sigma^{(1)}$; $V_{\perp}^{2\omega} \propto \mathbf{D}/(\sigma^{(1)})^2$ and $\sigma^{(2)} = -\sigma^{(1)} E_{\perp}^{2\omega}/(E_{\parallel}^{\omega})^2 \propto \mathbf{D}\sigma^{(1)}$. Given that $\sigma^{(1)}$ monotonically decreases with increasing temperature, the peak of $V_{\perp}^{2\omega}$ shifts to a higher temperature, whereas the peak of $\sigma^{(2)}$ shifts to a lower temperature. We have clarified the relationship between the peak temperatures of $V_{\perp}^{2\omega}$, $E_{\perp}^{2\omega}/(E_{\parallel}^{\omega})^2$ and $\sigma^{(2)}$ in the revised manuscript (page 6) and added the above discussions and Fig. R2 to the Supplementary materials (Note 9 and Supplementary Fig. S13)

Fig. R2: Temperature dependences of normalized $V_{\perp}^{2\omega}$, $E_{\perp}^{2\omega}/(E_{\parallel}^{\omega})^2$, $\sigma^{(1)}$ and $\sigma^{(2)}$.

In addition, for the DC current-DC voltage measurement, I wonder what is situation on the opposite side (i.e., $I_{dc} < 0$).

Response: For the DC current-induced NLHE measurement, due to the unavoidable misalignment of the electrical contacts, the measured DC signal contains two components, i.e. the linear response resulting from the longitudinal resistance and the quadratic response arising from the NLHE. Since these voltage components are, respectively, odd and even functions of the applied current, we can easily separate them via anti-symmetrizing and symmetrizing the measured data, i.e. $V^{linear}(I) = [V(+I) - V(-I)]/2$ and $V^{NLHE}(I) = [V(+I) + V(-I)]/2$. The V^{DC} shown in the manuscript is actually the $V^{NLHE}(I)$ which is already symmetrized. In Fig. R3 attached below, we show the raw data of the AC and DC I - V characteristics in the $-200 - 200 \mu\text{A}$ range for sample C3 at 300K. Since the negative AC current is meaningless, we use $V(-I) = -V(I)$ to extend the AC I - V curve into the full range. As seen in Fig. R3, since the AC second-harmonic response can be screened out by the lock-in amplifier, the $I^\omega - V^\omega$ curve is linear which follows Ohm's law. In contrast, for the $I_{Raw}^{DC} - V_{Raw}^{DC}$ curve, the linear and quadratic terms are mixed and cannot be separated by the nanovoltmeter. Therefore, the $I^{DC} - V^{DC}$ curve is a little bit deviated from Ohm's law. After symmetrizing the $I^{DC} - V^{DC}$ data, we eliminated the linear component and thus obtained the quadratic DC I - V from the NLHE as shown in Fig. 3d in the manuscript.

Fig. R3: DC I - V and first-order AC I - V characteristics for sample C3 at 300K

Otherwise, I found the paper is interesting, well-written and with sufficient references. I recommend its publication if the authors address my concerns above.

Response: We thank the referee for this consideration. We hope we have satisfactorily addressed the issue raised by the referee.

Response to Reviewer #3

Reviewer #3 (Remarks to the Author):

Min et al. reported observation of the nonlinear Hall effect in a candidate Dirac material BaMnSb₂. Second harmonic and rectified Hall voltages quadratic to the applied current were observed. The effect is found to be stronger when current is applied along the crystal b-axis, but much weaker when the current is along the crystal a-axis. These are solid evidence for the nonlinear Hall effect. However, the manuscript may not be suitable for publication in Nature Communications in its current form for the following reasons:

Response: We thank the referee for taking time to review our manuscript and finding our experimental evidence for the nonlinear Hall effect of BaMnSb₂ is solid. We also appreciate the referee's questions, which have been extremely helpful in improving the manuscript.

1, Nonlinear Hall effect has been reported in bulk materials, such as Weyl semimetals (ref. 33, 34). The observed nonlinear Hall effect here is also not that different from the strained transition metal dichalcogenides in Ref. 29, which are also massive Dirac materials. Moreover, the effect has been observed at high temperatures (ref. 14). These studies compromise the novelty of this work.

Response: While nonlinear Hall effect (NLHE) has been observed in bulk materials and massive Dirac materials at low temperatures, and in TaIrTe₄ at room temperature, our observed NLHE in BaMnSb₂ has novelty in several aspects as recognized by reviewers 1 and 2.

Firstly, the only established example of room temperature NLHE in TaIrTe₄ is just a surface effect as its higher bulk crystal symmetry prevents a bulk NLHE. In contrast, the room temperature NLHE in BaMnSb₂ is *the first example of bulk NLHE at room temperature*. Every Sb zig-zag chain layer in BaMnSb₂, which generates a spin-valley locked state, contributes to the NLHE. Compared to the surface NLHE, bulk NLHE could have much larger nonlinear Hall conductivity. As we have shown in our response to reviewer 1 (Fig. R1), the maximal 3D nonlinear Hall conductivity of our HB1 sample is ~22 S/V, which is several orders of magnitude larger than that of TaIrTe₄ (whose nonlinear conductivity is 0.01-0.04 S/V). Large nonlinear conductivity Hall conductivity is critical for applications since it can lead to large photocurrent. We have added this information to the revised manuscript (page 6 and Supplementary Fig. S13).

Secondly, as pointed out by referee 1, we demonstrate the wireless frequency doubling based on NLHE in the microwave range for the first time. Such a demonstration may facilitate NLHE's applications in technologies that require frequency doubling.

In addition, from the view of fundamental science, the mechanism of NLHE in BaMnSb₂ is also unique. Different from the 2D strained transition metal dichalcogenides, bulk BaMnSb₂ hosts a bulk spin-valley locked state. Our demonstration of the large room-temperature NLHE generated by the bulk spin-valley electronic state in BaMnSb₂ points to a direction for the search for bulk materials showing large room-temperature NLHE.

2, The authors claim that the observed nonlinear Hall effect is an intrinsic Berry curvature effect based on the non-monotonic temperature dependence. They claim the material is slightly electron-doped at low temperatures, and the carrier density increases with temperature due to thermal activations. However, this claim is questionable. First, the authors only showed carrier densities at temperatures of 10K and near the nonlinear Hall maximum. How does the carrier density change at higher temperatures? Is the carrier density change at higher temperatures consistent with the rapid reduction of the nonlinear Hall voltage? Second, the nonlinear Hall voltage switches sign as sample temperature varies, most evidently

in sample C4, at $T \sim 200\text{K}$. Such sign reversal in NLHE is not expected unless carrier type changes or band structure renormalizes.

Response: We thank the referee for carefully scrutinizing our data.

To answer the first question, we list the carrier densities at higher temperatures in Table. R1. The carrier densities of all the samples are indeed found to increase with further increasing the temperature above room temperature except for sample C2. This result is consistent with our calculations shown in Fig 4b in the main text, namely the higher temperature pushes the chemical potential upward, leading to the increase of the carrier density. We have updated Table S1 in the supplementary materials and added the carrier densities measured at 350K (or 325K)

Table R1: Carrier density n_e measured at various temperatures for samples C1, C2, C3, C4, HB1, and HB2.

Sample label	$n_e(10\text{K})$ (10^{19} cm^{-3})	$n_e(\text{RT})$ (10^{19} cm^{-3})	$n_e(350\text{K})$ (10^{19} cm^{-3})
C1-R	2.0	7.4	15
C2	2.4	2.4	2.4
C3	0.8	0.8	1.3
C4	3.2	8.2	20 (325k)
HB1	4.4	17	
HB2		2.4	3.0

The referee asked a very good question here regarding the sign change of NLHE reflected in our data, which motivated us to perform further data analyses. We have now better understood this characteristic. First, we would like to point out that the weak background nonlinear Hall signals away from the peaks are likely to be extrinsic (either from disorder scattering induced NLHE or other second-order signals resulting from sample asymmetry, thermoelectric effect, etc.). These weak extrinsic NLHE signals can have the same or opposite sign relative to the intrinsic NLH signal. However, the situation of sample C4 is essentially different. As we discussed in the manuscript, BaMnSb_2 crystals often show 180° domains in the c direction. The Berry curvature dipole rotates 180° across such a 180° domain wall, which leads the NLHE generated by such two domains to have opposite signs. Moreover, the inhomogeneity of the disorder distribution inside the sample can result in slightly different chemical potentials between different domains. Consequently, we can imagine a scenario where two kinds of c -axis domains produce opposite NLHE and their maximal values occur at slightly different temperatures due to chemical inhomogeneity, thus resulting in a sign change in the NLH voltage with the variation of temperature.

As shown in Fig. R4 attached below, we can fit the data of sample C4 with this model. Our fitted data clearly show the 2nd harmonic Hall voltage peak of sample C4 can be expressed as a sum of one positive and one negative Gaussian peaks with the same width which respectively represent the NLHE generated in two different kinds of domains. The curve of domain 1 shows a peak at 273K, while the curve of domain 2 shows a peak at 267K. The sum of these two peaks agrees well with the raw data. This result also explains why the peak of sample C4 is narrower compared to the peaks of other samples, as reflected in Fig. 4C in the manuscript. If we plot the temperature-dependent NLH voltage of only one domain together with those of the other samples, we find they all have similar peak profiles.

Of course, other samples also contain 180° domains as well. However, as long as the chemical potentials of these domains are close to each other, the overall temperature-dependent NLH signal will still be in single-peak shape without a significant sign change. As the samples used in our experiments were small, for most of the samples, the chemical inhomogeneity was not severe. That is why only sample C4 shows an obvious sign change in NLHE. We have added this paragraph of discussion to the main text (page 6 and Supplementary Note 3) and supplementary materials (Supplementary Note & Fig. S10).

Fig. R4: Two-peak fitting of the temperature-dependent NLH voltage measured on sample C4.

We hope we have satisfactorily addressed all the issues raised by the referee.

REVIEWERS' COMMENTS

Reviewer #1 (Remarks to the Author):

The authors have addressed my comments properly. The paper is well written. The novelty the study certainly meet the standards of Nature Communications. I recommend the publication of the manuscript in Nature Communications.

Reviewer #2 (Remarks to the Author):

The authors satisfactorily addressed my questions from the last round. I support its publication now.

Reviewer #3 (Remarks to the Author):

The authors have addressed my concerns on the novelty of this study. I thank the authors for pointing out the unique aspect of this material; I appreciate the system better now; and I have no concern on the novelty.

The authors also attempted to address the sign change in the nonlinear Hall response with temperature using a crystal domain model. While I am a bit skeptical on this interpretation, I find it sufficient to leave the paper like this at this point. Further studies are required to dig out the details on the sign change in certain samples.

I recommend publication.

Response Letter

We would like to thank all the reviewers for kindly reviewing our paper and the editors for pointing out the editorial issues. In the following, we provide a point-to-point response (shown in blue) to all comments.

Response to Reviewer #1

Reviewer #1 (Remarks to the Author):

The authors have addressed my comments properly. The paper is well written. The novelty the study certainly meet the standards of Nature Communications. I recommend the publication of the manuscript in Nature Communications.

Response: We thank the referee for the positive comments. We are delighted to hear that our revised paper meets the standards of Nature Communications.

Response to Reviewer #2

Reviewer #2 (Remarks to the Author):

The authors satisfactorily addressed my questions from the last round. I support its publication now.

Response: We thank the referee for kindly reviewing our manuscript. We are glad to share our results through the publication in Nature Communications.

Response to Reviewer #3

Reviewer #3 (Remarks to the Author):

The authors have addressed my concerns on the novelty of this study. I thank the authors for pointing out the unique aspect of this material; I appreciate the system better now; and I have no concern on the novelty.

The authors also attempted to address the sign change in the nonlinear Hall response with temperature using a crystal domain model. While I am a bit skeptical on this interpretation, I find it sufficient to leave the paper like this at this point. Further studies are required to dig out the details on the sign change in certain samples.

I recommend publication.

Response: We thank the referee for taking time to review our manuscript and appreciate the positive judgment on the novelty of our work.

As the referee is still skeptical about our interpretation of the sign change in the nonlinear Hall response versus the temperature curve, we have modified our statement from “This can be well understood in

terms of 180° domains and slight chemical inhomogeneity” to “This can be possibly interpreted by 180° domains and slight chemical inhomogeneity”.

We are delighted that the referee recommended our work to be published in Nature Communications.